# Organizational Benefits of Ultra-Low-Dose Chest CT Compared to Chest Radiography in the Emergency Department for the Diagnostic Workup of Community-Acquired Pneumonia: A Real-Life Retrospective Analysis

**DOI:** 10.3390/medicina59091508

**Published:** 2023-08-22

**Authors:** Sabrina Kepka, Charlène Heimann, François Severac, Louise Hoffbeck, Pierrick Le Borgne, Eric Bayle, Yvon Ruch, Joris Muller, Catherine Roy, Erik André Sauleau, Emmanuel Andres, Mickaël Ohana, Pascal Bilbault

**Affiliations:** 1Emergency Department, Hôpitaux Universitaires de Strasbourg, 1 Place de l’Hôpital, CHRU of Strasbourg, 67091 Strasbourg, France; louise.hoffbeck@chru-strasbourg.fr (L.H.); pierrick.leborgne@chru-strasbourg.fr (P.L.B.); eric.bayle@chru-strasbourg.fr (E.B.); pascal.bilbault@chru-strasbourg.fr (P.B.); 2ICUBE, UMR 7357, CNRS, 300 Bd Sébastien Brant, 67400 Illkirch-Graffenstaden, France; francois.severac@chru-strasbourg.fr (F.S.); erik.sauleau@chru-strasbourg.fr (E.A.S.); 3Emergency Department, Hôpital Emile Muller, 20 rue du Dr Laennec, 68100 Mulhouse, France; charlene.heimann@gmail.com; 4Méthodes en Recherche Clinique (GMRC), Hôpitaux Universitaires de Strasbourg, 1 Place de l’Hôpital, 67091 Strasbourg, France; 5UMR 1260, INSERM/Université de Strasbourg CRBS, 1 Rue Eugene Boeckel, 67000 Strasbourg, France; 6Department of Infectious and Tropical Diseases, Hôpitaux Universitaires de Strasbourg, 1 Place de l’Hôpital, 67091 Strasbourg, France; yvon.ruch@chru-strasbourg.fr; 7Public Health Units, Hôpitaux Universitaires de Strasbourg, 1 Place de l’Hôpital, CHRU of Strasbourg, 67091 Strasbourg, France; joris.muller@chru-strasbourg.fr; 8Radiology Department, Nouvel Hôpital Civil, Hôpitaux Universitaires de Strasbourg, 1 Place de l’Hôpital, 67091 Strasbourg, France; catherine.roy@chru-strasbourg.fr (C.R.); mickael.ohana@chru-strasbourg.fr (M.O.); 9Department of Internal Medicine, Hôpitaux Universitaires de Strasbourg, 1 Place de l’Hôpital, 67091 Strasbourg, France; emmanuel.andres@chru-strasbourg.fr

**Keywords:** community-acquired pneumonia, ultra-low-dose chest computed tomography, chest radiography, emergency department

## Abstract

*Background and Objectives:* Chest radiography remains the most frequently used examination in emergency departments (ED) for the diagnosis of community-acquired pneumonia (CAP), despite its poor diagnostic accuracy compared with ultra-low-dose (ULD) chest computed tomography (CT). However, although ULD CT appears to be an attractive alternative to radiography, its organizational impact in ED remains unknown. Our objective was to compare the relevant timepoints in ED management of CT and chest radiography. *Materials and Methods:* We conducted a retrospective study in two ED of a University Hospital including consecutive patients consulting for a CAP between 1 March 2019 and 29 February 2020 to assess the organizational benefits of ULD chest CT and chest radiography (length of stay (LOS) in the ED, time of clinical decision after imaging). Overlap weights (OW) were used to reduce covariate imbalance between groups. *Results:* Chest radiography was performed for 1476 patients (mean age: 76 years [63; 86]; 55% men) and ULD chest CT for 133 patients (mean age: 71 [57; 83]; 53% men). In the weighted population with OW, ULD chest CT did not significantly alter the ED LOS compared with chest radiography (11.7 to 12.2; MR 0.96 [0.85; 1.09]), although it did significantly reduce clinical decision time (6.9 and 9.5 h; MR 0.73 [0.59; 0.89]). *Conclusion:* There is real-life evidence that a strategy with ULD chest CT can be considered to be a relevant approach to replace chest radiography as part of the diagnostic workup for CAP in the ED without increasing ED LOS.

## 1. Introduction

Community-acquired pneumonia (CAP) is one of the most frequent causes of hospital admissions, accounting for one million hospital admissions per year in Europe [1,2,3]. Incidence ranges from 1.5 to 14 cases per 1000 person-years [4,5,6,7], and some studies revealed that it is around four times higher in hospitalized patients over 75 years of age than in those under 65 [7,8]. Indeed, the elderly are more often affected, and the risk of death, up to 48%, correlates with increasing age [9]. CAP is therefore a major public health problem, being the most frequent cause of death from infection in Europe and the United States [2].

CAP represents a clinical concern in the emergency departments (ED), as 75% of hospitalized patients are admitted via these units [10]. Chest imaging is therefore recommended to establish an accurate diagnosis in admitted patients, to enable appropriate management and early treatment [11]. Despite the low diagnostic accuracy of chest radiography for CAP compared to chest computed tomography (CT), it remains the most frequently used form of imaging for diagnostic workup in the ED, with significant consequences for the management of these patients. Indeed, chest radiography is more likely than CT to lead to overdiagnosis and overuse of antibiotics [12]. On the other hand, under-diagnosis is more frequent, which delays treatment and can have serious consequences [13]. Early chest CT in the ED is a more accurate technique and reduces antibiotic prescription [14]. However, radiation exposure and higher costs may limit the use of chest CT in the ED [15,16].

Ultra-low-dose (ULD) CT offers reduced radiation exposure, comparable with chest radiography, and better diagnosis accuracy despite lower image quality to standard chest CT. Indeed, a previous study revealed that the probability of pneumonia was altered in 45% of elderly patients by ULD chest CT compared with chest radiography [17]. Nevertheless, CT may lead to an increase in length of stay (LOS) in the ED [18]. However, the organizational benefits of a thoracic ULD CT strategy in the ED have not been previously evaluated, even though this is a major factor in the implementation of this imaging in the diagnostic work-up of CAP. In addition to organization in these units, a prolonged LOS in the ED could have several adverse effects for patients, including higher mortality [19,20,21], which could limit the use of ULD chest CT in the ED despite better diagnostic accuracy.

Our objective was therefore to assess the organizational benefits of the imaging strategy for the management of CAP on the ED care pathway, by comparing LOS and time to clinical decision making in the ED for both strategies.

## 2. Materials and Methods

### 2.1. Study Design

We conducted a retrospective study in an ED of a university hospital in France from 1 March 2019 to 29 February 2020.

### 2.2. Study Setting and Population

We included all consecutive patients in the ED aged over 18 years consulting for a CAP and with a chest radiography or a ULD chest CT performed in the ED for diagnostic workup (Figure 1). Patients with a type of chest CT other than ULD chest CT were excluded from this study.

Chest imaging was performed according to the advice of emergency physicians and radiologists under real-life conditions with no criteria for performing either imaging. After thoracic imaging, patient management and orientation were carried out by emergency physicians on the basis of imaging results, clinical presentation, and laboratory tests.

### 2.3. Endpoints

The main endpoint was to evaluate the organizational benefits of ULD chest CT and chest radiography on the ED care pathway for patients with CAP, including:Length of stay (LOS) in the ED;Time from imaging to discharge;Overall LOS in the ED.

LOS was measured as the time from ED admission to ED discharge (discharge or transfer to a medical unit), as recorded in the ED’s electronic health record. Time to clinical decision was measured as time from imaging to ED discharge; overall ED length of stay included time required for ED management and time spent in the ED short-stay unit, and was recorded in the ED’s electronic health record.

For the secondary endpoints, we measured:Agreement between ED diagnosis and hospital discharge diagnosis

Inpatient discharge diagnosis was taken as the reference standard. This analysis covered only inpatients in the cohort. In addition, a subgroup analysis was performed for patients with chronic heart failure.

Factors associated with optimal antibiotic prescribing

The effect of imaging on the agreement between prescription in the ED and during hospitalization was assessed in a multivariate analysis for the cohort’s hospitalized patients and in the subgroup of patients with chronic heart failure.

### 2.4. Measurements

All patients presenting with CAP (International Classification of Diseases (ICD) diagnostic code J18 and J15) during the study period were identified retrospectively using medical informatics queries. ED and in-hospital surveillance data were obtained from local hospital databases. Age, gender, and medical history; clinical parameters (blood pressure, oxygen saturation, temperature, and heart rate); and biological data were recorded. Radiological data collected in the ED were also classified as unilateral or bilateral pneumonia, bronchiolitis, or signs of heart failure.

### 2.5. Statistical Analysis

Quantitative variables were described as the mean ± standard deviation or median with first and third quartiles according to the normality of the distribution, while categorical variables were described as numbers and percentages. In order to compare the outcomes between the two types of imaging (ULD chest CT or chest radiography) taking into account potential confounders, we performed propensity score weighting using overlap weights (OW). The most common stabilized weights (SW) were also used to perform a sensitivity analysis. This allowed us to create a pseudo-population (weighted population) in which patients receiving both types of imaging have similar characteristics. We used absolute standardized mean differences (SMD) to assess the comparability of the baseline covariates between the two groups. SMD values close to 0 indicate insignificant differences between groups. Marginal structural models were then used to compare outcomes between the groups in the weighted sample. Gamma regression models were performed for continuous variables. Results are presented as means ratios (MR) with a 95% confidence interval (CI). Logistic regression models were used for binary variables, and the results are presented as odds ratios (OR) with 95% CI.

We performed a logistic regression to compare the rate of diagnostic agreement between the two groups (chest radiography and ULD chest CT) and to assess the effect of imaging strategy on diagnostic agreement between ED discharge and hospitalization discharge on a weighted population. The OR represented the strength of the association between the imaging strategy and the probability of agreement. Furthermore, factors associated with an optimal prescription of antibiotics in the ED were identified using a multivariable logistic regression model. All the variables presenting a clinical relevance or a *p*-value < 0.2 in univariable analysis were included into the multivariable model.

The datasets employed for the main analysis was the entire study population. Subgroup analyses were conducted for diagnostic agreement and antibiotics agreement between ED and discharge, including only patients with a history of heart failure. Statistical significance was set at *p* < 0.05. Analyses were performed using R Version R 4.0.3 (R Foundation for Statistical Computing, Vienna, Austria). A more detailed description of the statistical analyses is available in Appendix A.

### 2.6. Ethics Approval, Data, and Safety Monitoring

This study was conducted in accordance with the principles set forth by Good Clinical Practice guidelines and the Declaration of Helsinki. The study was approved by the Ethics Committee (CE 2021-50). A declaration of conformity was obtained from the Commission nationale de l’informatique et des libertés (CNIL) (agreement number 2208067v0). In accordance with French legislation, formal written informed consent was not required for this type of study because data were entirely retrospectively studied [22].

## 3. Results

We included a total of 1609 consecutive patients consulting in the ED for a CAP with chest radiography or ULD chest CT performed in the ED (Figure 1). The majority (N = 1476) underwent chest radiography, and a ULD chest CT was performed in 133 patients (Figure 1). The median age was 76 years [63; 86] for patients with chest radiography and 71 years [57; 83] for patients with ULD chest CT (SMD = 0.20) (Table 1).

No difference was highlighted between the groups regarding medical history and clinical presentation.

More abnormalities compatible with pneumonia (SMD = 0.36) and bronchiolitis (SMD = 0.62) were highlighted for chest CT than for chest radiography.

The cohort characteristics and SMD in the weighted population with overlap weights are presented in Appendix B (Table A1).

The balance of covariates between groups was greatly improved in the weighted populations, with SMD < 0.2 for all baseline characteristics, making the groups comparable with regard to the variables included in the propensity score in the weighted population, whatever the type of weighting (Figure 2). Comparability between groups appears to be better with the overlapping weight (OW) than with the stabilized weight, also used to perform a sensitivity analysis. The results are presented for the weighted population (OW).

### 3.1. Main Endpoint: Organizational Benefits by Assessing the Impact of the Imaging Strategy on ED Care Pathway for the Diagnostic Workup of CAP

*The mean ED LOS* was 11.7 h for patients who underwent ULD chest CT and 11.9 h for patients who underwent chest radiography. There was no difference in ED LOS between ULD chest CT and chest radiography, with a mean ratio of 0.96 [95% CI 0.85; 1.09] in the weighted population with OW (Table 2).

Sensibility analysis in the unweighted population and in the weighted population with stabilized weights is presented in Appendix C (Table A2).

*The mean time from imaging to ED discharge* was significantly shorter for ULD chest CT (7.1 h) than for chest radiography (9.2 h), with a mean ratio of 0.73 [95% CI 0.59; 0.89] for the weighted population with OW (Table 2).

Finally, *the mean global LOS in the ED*, including time spent in the ED and time spent in the emergency short-stay unit, was 23.3 h for patients who underwent chest CT in the ULD and 23.9 h for chest radiography, with no difference between groups (mean ratio of 1.05 [95% CI 0.87; 1.26] in the OW-weighted population) (Table 2).

### 3.2. Secondary Endpoints

#### 3.2.1. Diagnosis Agreement

In the unweighted population, image findings were concordant with discharge findings for 838 (68.7%) patients and for 103 (88%) patients in the radiography group and in ULD CT group, respectively (*p* < 0.001). Agreement between diagnosis in the ED and diagnosis at hospital discharge for inpatients was significantly higher in the group with ULD chest CT than in the group with chest radiography, which were 87% and 68%, respectively, with an OR 1.92 [95% CI,1.04;3.57] in the weighted population with OW (Table 3).

Sensibility analysis in the unweighted population and in the weighted population with stabilized weights are presented in Appendix D (Table A3).

For the group displaying signs of chronic heart failure, diagnosis agreement was performed for 100% of patients with ULD chest CT and 73% with chest radiography (*p* = 0.003 in the weighted population with OW).

#### 3.2.2. Antibiotics Agreement between ED and Discharge

We showed in the multivariable model that a C reactive protein (CRP) level higher than 100 mg/L and a prescription of antibiotics in the ED were associated with an agreement between antibiotic prescription in the ED and antibiotic prescription during hospitalization, with an OR of 0.75 [95% CI 0.58–0.97] and 1.95 [95% CI 1.27–2.99], respectively (Figure 3).

In the chronic heart failure group, the factors associated with an agreement between antibiotic prescription in the ED and referral (hospitalization) were being male, having a fever, and having a CRP level above 100 mg/L, with an OR of 2.12 [95% CI 1.02–4.42], 2.34 [95% CI 1.06–5.17], and 0.33 [95% CI 0.16–0.71], respectively (Figure 4).

No association was found between ULD CT in the ED and antibiotic prescription agreement in the cohort and in the subgroup with chronic heart failure.

## 4. Discussion

In our study, LOS was not increased by ULD chest CT for the diagnostic workup of CAP in the ED compared with chest radiography, at 11.7 h and 11.9 h, respectively. Time from imaging to discharge from ED was significantly reduced with ULD chest CT compared with chest radiography, at 7.1 and 9.2 h, respectively. Furthermore, agreement between ED diagnosis and discharge diagnosis for inpatients was significantly higher in the ULD chest CT group than in the chest radiography group (OR 1.92 [1.04;3.57] for the OW-weighted population) and in the subgroup of patients with chronic heart failure. No correlation was found between ULD CT in the ED and agreement between antibiotic prescription in the cohort and in the subgroup with chronic heart failure.

CAP is more common in elderly patients, with an average age of 62 years in a study conducted in the EDs of three academic hospitals in the United States [23]. Cough and dyspnea remain the most common symptoms [15]. However, CAP is difficult to diagnose in the ED due to non-specific symptoms in around 14% of cases, notably in elderly patients and patients with a history of congestive heart failure [15,23]. A previous study concluded that clinical examination was not discriminant in detecting CAP in the ED, since the positive predictive value of symptoms in the ED for elderly patients was 0.16 [95% CI 0.12; 0.19] [24].

LOS in the ED is crucial for early and appropriate treatment. Few studies have reported on the organizational consequences of different imaging examinations in the ED. To our knowledge, only one study has reported ED LOS for CAP in France [25], which was 326 min [95% CI 268; 359]. In other countries, it ranged from four to six hours (240 to 360 min) [25]. This LOS is shorter than that found in our study, which could be explained by the high proportion of patients at our center with cardiovascular and pulmonary disease requiring longer management. Furthermore, this study concerned patients between March and October 2013, and ED LOS has increased significantly since, due to an increase in ED admissions. However, reducing LOS is a priority in these units, as ED overcrowding is associated with increased hospital LOS and mortality [26,27,28]. Prolonged ED LOS by chest CT could be a limitation to the implementation of ULD chest CT in the ED, and our results confirm that systematic ULD chest CT could replace chest radiography in the ED for the diagnostic workup of CAP without negative organizational consequences. Furthermore, ED diagnosis often differs from discharge diagnosis for the management of CAP [23]. To improve diagnosis accuracy in the ED, CT, and especially ULD chest CT, is a more accurate alternative to chest radiography for CAP [14,17]. The availability of CT in the ED is a limitation for some authors, and a CAP diagnostic algorithm has been used to identify the patients that who might benefit from CT in the ED [15]. However, the availability of CT in the ED is increasing over time. Lung ultrasound is another imaging strategy that can help emergency clinicians diagnose CAP with a better accuracy compared with chest radiography, using chest CT scan as the gold standard (the Z statistic was 3.093 (*p* = 0.002), and the areas under the curve for ultrasound and chest radiography were 0.901 and 0.590, respectively) [29]. Then, using hospital discharge diagnosis as the reference standard, the calculated pooled sensitivity for lung ultrasound is 0.95 (0.93–0.97) [29]. Moreover, lung ultrasound could help physicians monitor the therapeutic effect after an initial ULD CT in the ED, enabling easier comparative assessment than chest radiography.

In addition, reducing time to antibiotic administration is essential for the appropriate management of CAP in the ED, and depends on the chosen imaging strategy. Indeed, in a study conducted in the USA, ninety percent of antibiotics were prescribed in the ED, and patients who received antibiotics more than four hours after ED arrival experienced longer waits for radiograph orders [30]. A study conducted in three hospitals in Switzerland, the United States, and France revealed that time to antibiotics was 310 min for CAP (220–359) [25]. A delay in antibiotics for patients admitted with pneumonia occurred more frequently for patients with non-classic symptoms or with chronic heart failure, which corresponds to a high proportion of patients in the ED with a suspicion of CAP [30]. Indeed, delays in ED antibiotic initiation are associated with higher mortality from sepsis; each additional hour from ED arrival to antibiotic initiation for sepsis is associated with a 10% increased odds of 1-year mortality (95% CI, 5–14; *p* < 0.01) [31]. In addition to avoiding adverse events, an adequate prescription of antibiotics is necessary to reduce antibiotics consumption and bacterial resistance. For patients with moderately severe CAP requiring hospitalization with clinical stability after 3 days of β-lactam therapy, a discontinuation strategy if antibiotic treatment proves not to be less than 8 days of treatment supports the reduction of antibiotic consumption [32]. In our study, the increased delay in access to imaging in the ULD CT group can have had an impact on the earliness of antibiotic prescription. However, imaging is not the only factor taken into account for antibiotic prescription. The implementation of measures to reduce treatment times and reduce ED stay is a way to improve outcomes for these patients.

ULD chest CT became a key to ruling out the diagnosis of CAP with a better accuracy than chest radiography and to reduce the inappropriate prescription of antibiotics without an increase in LOS in the ED [14,17,33]. Then, ULD chest CT could allow a significant saving of antibiotics in the context of increasing bacterial resistance.

### Limitations and Strengths

Despite our retrospective design and the lack of randomization, which could limit the impact of this study, we performed an inverse probability weighting from propensity scores to take into account the heterogeneity between the groups. Furthermore, time from imaging to discharge could have been considered as a surrogate endpoint of time to clinical decision, and it could be impacted by others factors, such as hospital overcrowding. However, this time seemed the most relevant to approximate the time to clinical decision in this retrospective study.

In addition, few ULD chest CTs were performed in the ED due to the onset of the COVID-19 outbreak in 2020, which prevented patients from being included after February 2020, when the first patient was diagnosed in our center. However, the current pandemic has revealed the efficacy of chest CT in the ED in allowing a rapid and accurate diagnosis of COVID-19. Furthermore, the implementation of ULD chest CT was early, and the evaluation of organizational benefits would likely require more time to consider other implementation factors and corrective measures.

Finally, diagnostic agreement concerned only inpatients, as we considered the diagnosis at discharge for inpatients as the reference standard and compared it with the diagnosis at discharge in the ED before the transfer to a medical ward, implying some limits as it could have been influenced by other factors. While the use of discharge diagnosis as the “reference standard” could be debated, the superior diagnostic accuracy of CT is no longer in doubt. Thus, the aim of this pragmatic real-life study was to assess the organizational impact and to find out whether the management of patients was improved by ultra-low-dose CT. Thus, we considered this to be the best way to evaluate diagnostic agreement in this retrospective study, and it appeared interesting to evaluate whether imaging performed in the ED could modify the management in the ED when considering the hospital discharge as the reference.

Despite these limitations, our results reveal the effectiveness of the ULD chest CT performed in the ED compared with chest radiography for the diagnosis of CAP. To our knowledge, it is the first study in real-life conditions to focus on the organizational benefits of the imaging strategies performed in the ED for the diagnostic workup of patients presenting with CAP. It is important to emphasize the relevance of CT for differential diagnoses, especially in people of age or with risk factors, who represent a large proportion of patients consulting the ED. ULD chest CT enabled efficient triage with organizational benefits by improving orientation after ED (discharge or transfer to a relevant medical ward) to optimize the flow of ED patients, ensuring that patients spend less time in the ED. However, the need for radiologists trained in the interpretation of ULD CT and the availability of CT facilities may be seen as limitations if this modality were to become widespread, particularly given the impact on triage and treatment time if ULD CT are performed much later than chest radiography. The reduction in access time to imaging is a priority, and is being addressed by joint procedures involving radiologists and emergency physicians in order to improve the fluidity of emergency care pathways. The implementation of clinical pathways to reduce ED time with rapid access to ULD chest CT is a challenge for radiologists and ED physicians to overcome, in order to allow a shorter time to adequate treatment and to decrease the possibility of overcrowding in these units.

## 5. Conclusions

There is real-life evidence that a strategy with ULD chest CT can be considered as a relevant approach to CAP in the ED, improving patient care without increasing LOS in the ED. Rapid access to ULD chest CT should be considered to replace chest radiography and improve the ED management of CAP.

## Figures and Tables

**Figure 1 medicina-59-01508-f001:**
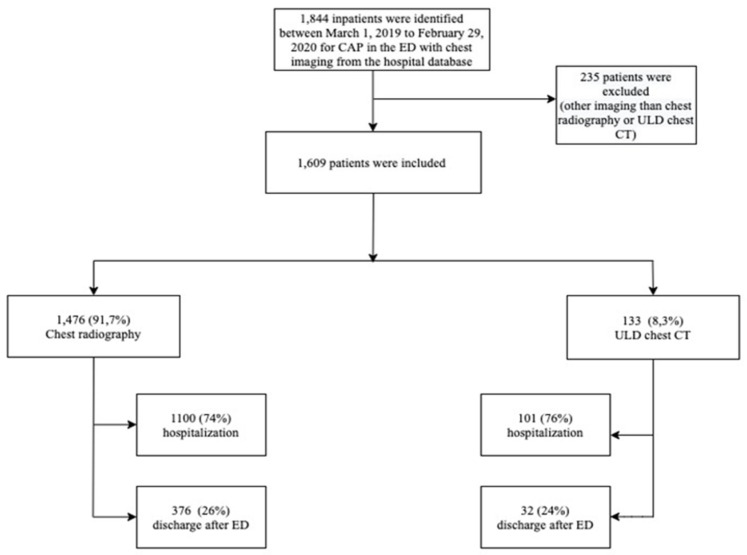
Flowchart of the study concerning the imaging diagnostic workup of CAP conducted in the ED of a university hospital in France. CAP, community-acquired pneumonia; CT, computed tomography; ED, emergency departments; ULD, ultra-low dose.

**Figure 2 medicina-59-01508-f002:**
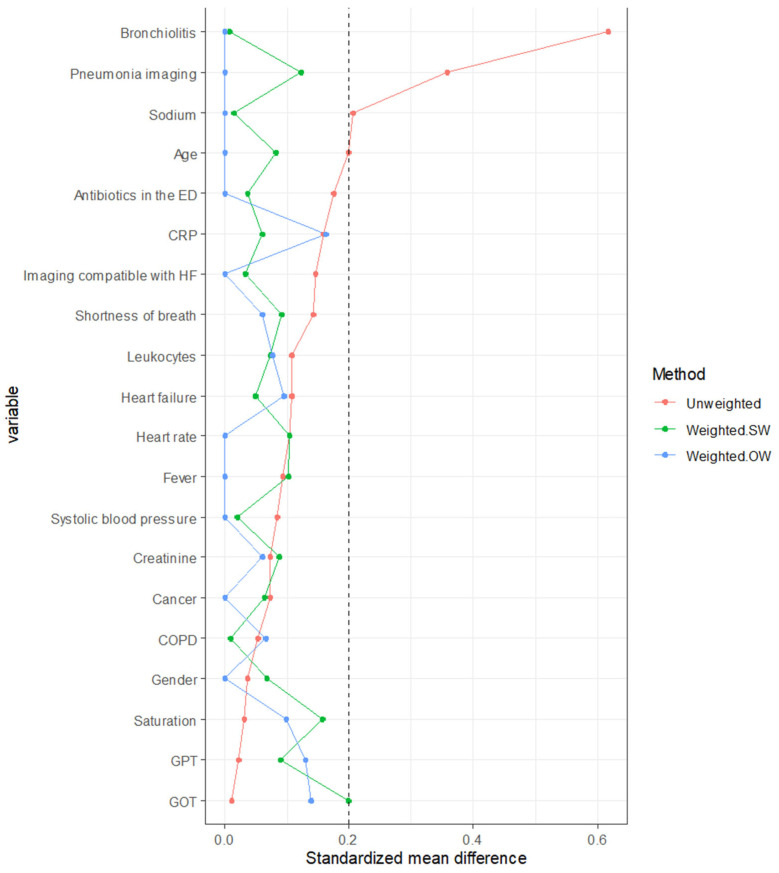
Standardized mean difference in weighted and unweighted populations. Weighted SW, weighted with stabilized weights; weighted OW, adjusted with overlap weights; CRP, C reactive protein; HF, heart failure; GOT, glutamic oxaloacetic transaminase; GPT, glutamic pyruvic transaminase.

**Figure 3 medicina-59-01508-f003:**
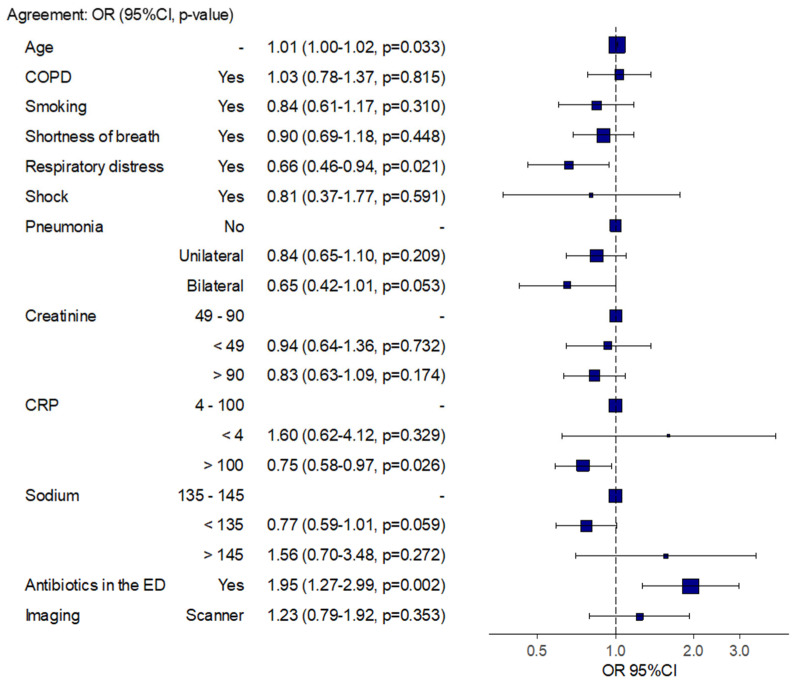
Agreement between antibiotics prescription in the ED and during hospitalization in the study population.

**Figure 4 medicina-59-01508-f004:**
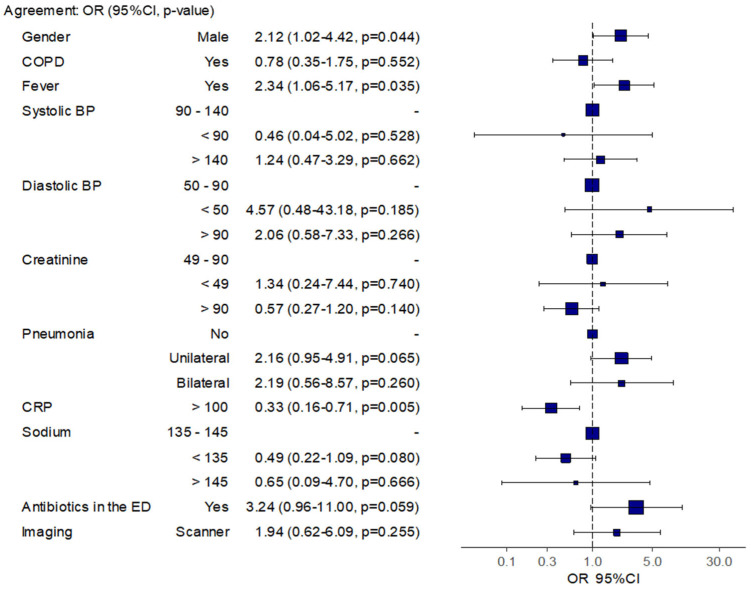
Agreement between antibiotics prescription in the ED and during hospitalization in the subgroup of patients with chronic heart failure.

**Table 1 medicina-59-01508-t001:** Cohort characteristics.

Parameters		Chest Radiography N = 1476	ULD Chest CT N = 133	*p*	SMD
Age (median, quartiles)		76.0 [63.0, 86.0]	71.0 [57.0, 83.0]	**0.01**	**0.20**
Gender					
	Men (N, %)	815 (55.2)	71 (53.4)	0.75	0.04
	Women (N, %)	661 (44.8)	62 (46.6)		
Medical history					
	Heart failure (N, %)	188 (12.7)	22 (16.5)	0.26	0.11
	COPD (N, %)	389 (26.4)	32 (24.1)	0.63	0.05
	Cancer (N, %)	310 (21.0)	32 (24.1)	0.47	0.07
	Smoking habit (N, %)	281 (19)	28 (21.1)	0.65	0.05
Clinical presentation					
	Saturation (median, quartiles)	95.0 [93.0, 97.0]	95.0 [93.0, 97.0]	0.96	0.03
	Heart rate (M, SD)	95 (±20.4)	93 (±19.8)	0.26	0.10
	Systolic blood pressure (M, SD)	130 (±22.0)	132 (±21.1)	0.58	0.08
	Diastolic blood pressure (M, SD)	76.14 (±15.8)	80.33 (1±3.1)	**0.004**	**0.29**
	Septic shock (N, %)	34 (2.3)	2 (1.5)	0.77	0.06
	Shortness of breath (N, %)	761 (51.6)	78 (59)	0.14	0.14
	Respiratory failure (N, %)	185 (12.5)	14 (10.5)	0.59	0.06
	Fever (N, %)	567 (38.6)	45 (34.1)	0.36	0.09
Laboratory tests					
	Leukocytes (G/L) (median, quartiles)	12.1 [10.0, 14.0]	12.3 [10.3, 13.8]	0.61	0.11
	Creatinine (µg/L) (median, quartiles)	75.0 [57.0, 103.0]	71.4 [55.0, 90.5]	0.23	0.07
	CRP (mg/L) (median, quartiles)	73.0 [29.0, 150.0]	87.0 [29.0, 191.0]	0.24	0.16
	GOT (U/L) (median, quartiles	24.0 [17.0, 41.0]	26.5 [20.2, 37.5]	0.17	0.01
	GPT (U/L) (median, quartiles)	23.0 [17.0, 35.0]	25.0 [15.0, 39.5]	0.70	0.02
	Sodium (mmol/L) (median, quartiles)	137.0 [134.0, 139.0]	136.0 [134.0, 138.0]	**0.02**	**0.21**
Imaging results	Pneumonia				
	Absence of pneumonia	643 (43)	46 (34.6)	**<0.001**	**0.36**
	Bilateral pneumonia	106 (7.2)	25 (18.8)		
	Unilateral pneumonia	735 (49.8)	62 (46.6)		
	Bronchiolitis	32 (2.2)	28 (21.1)	**<0.001**	**0.62**
	Imaging compatible with heart failure	119 (8.1)	6 (4.5)	0.19	0.15
Antibiotics in the ED (N, %)		1308 (88.8)	110 (82.7)	**0.05**	**0.18**

ED, emergency department; COPD, chronic obstructive pulmonary disease; CRP, C reactive protein; GOT, glutamic oxaloacetic transaminase; GPT, glutamic pyruvic transaminase; M, mean; SD, standard deviation; SMD, standard mean difference.

**Table 2 medicina-59-01508-t002:** Comparison of length of stay in the ED between ULD chest CT and chest radiography in the weighted populations.

Parameters	Population	Chest Radiography	ULD Chest CT	MR	95% CI	*p*
		N = 1476	N = 133			
Time from imaging to ED discharge Mean (SD)	Weighted population (OW)	9.5 (7.7)	6.9 (7.3)	0,73	[0.59; 0.89]	0.002
ED length of stay Mean (SD)	Weighted population (OW)	12.2 (7.7)	11.7 (7.6)	0.96	[0.85; 1.09]	0.57
Global length of stay in the ED * Mean (SD)	Weighted population (OW)	22.4 (19.8)	23.4 (22.3)	1.05	[0.87; 1.26]	0.62

* Global length of stay in the ED including ED short-stay unit. ED, emergency department; MR means ratio CI; OW, overlap weights.

**Table 3 medicina-59-01508-t003:** Diagnosis agreement between ED discharge and hospitalization discharge, considered as the reference.

Parameter		Chest Radiography N = 1476	ULD Chest CT N = 133	OR	95% CI	*p*
**Diagnosis Agreement** **N (%)**	Weighted population (OW)	62.5 (78.0)	76.0 (87.1)	1.92	[1.04; 3.57]	**0.04**

OR, odds ratio CI; SW, stabilized weights; OW, overlap weights.

## Data Availability

In accordance with French legislation, the datasets are not available.

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
