# Peer review of "Organizational Benefits of Ultra-Low-Dose Chest CT Compared to Chest Radiography in the Emergency Department for the Diagnostic Workup of Community-Acquired Pneumonia: A Real-Life Retrospective Analysis"

_medicina, 2023, doi:10.3390/medicina59091508_

Round 1
Reviewer 1 Report
The article "Organizational benefits of ultra-low-dose chest CT compared to chest radiography in the emergency department for the diagnostic workup of community-acquired pneumonia: a real-life retrospective analysis." is a retrospective cohort study comparing the clinical benefits between plain chest film and LDLC. The author used the overlap weighted method to compare two groups with a significant case number difference. The results showed that ULD lung CT did not significantly increase the length of ED stay. Also, ULD lung CT provided better diagnostic agreement. The following are the critical weaknesses of the study.
1. ULD lung CT reduced only time after the image examination, which suggested that it might take longer for the patient to receive CT than X-ray. How does it affect the timing of antibiotic prescriptions?
2. ULD lung CT should provide better sensitivity and specificity in diagnosing pneumonia than X-ray. The author uses discharge diagnosis as the "gold standard," which could be problematic.
3. Although CT is available in most emergency departments, the cost-effective benefit should also be considered. Is there a financial analysis available in this database?
4. How are the image results different from the discharge results?
5. The writing needs improvement. (abstract line 29-32)
Author Response
Reviewer 1
The article "Organizational benefits of ultra-low-dose chest CT compared to chest radiography in the emergency department for the diagnostic workup of community-acquired pneumonia: a real-life retrospective analysis." is a retrospective cohort study comparing the clinical benefits between plain chest film and LDLC. The author used the overlap weighted method to compare two groups with a significant case number difference. The results showed that ULD lung CT did not significantly increase the length of ED stay. Also, ULD lung CT provided better diagnostic agreement. The following are the critical weaknesses of the study.
- ULD lung CT reduced only time after the image examination, which suggested that it might take longer for the patient to receive CT than X-ray. How does it affect the timing of antibiotic prescriptions?
Response: We agree with the reviewer. Longer imaging times may have had an impact on the earliness of antibiotic prescription. We agree that this is a limitation of the study. We have added it in the discussion as follows line 326:
“In our study, the increased delay in access to imaging in the ULD CT group can have had an impact on the earliness of antibiotic prescription. However, imaging is not the only factor taken into account for antibiotic prescription.”
- ULD lung CT should provide better sensitivity and specificity in diagnosing pneumonia than X-ray. The author uses discharge diagnosis as the "gold standard," which could be problematic.
Response: Once again, we agree with the reviewer and thank him for this pertinent comment. In fact, we don't have a reference CT to compare radiography and CT or CT and ultra-low-dose CT. However, the superior diagnostic accuracy of CT is no longer in doubt. Thus, the aim of this pragmatic real-life study was to assess the organizational impact and to find out whether the management of patients was improved by ultra-low-dose CT. We therefore took as a reference the diagnosis at discharge from hospital, as an expert opinion, even if this has its limitations. We specified this in the manuscript line 353.
“Finally, diagnostic agreement concerned only inpatients, as we considered the diagnosis at discharge for inpatients as the reference and compared it with the diagnosis at discharge in the ED before the transfer to a medical ward implying some limits as it could have been influenced by other factors. While the use of discharge diagnosis as the "gold standard” could be debate, the superior diagnostic accuracy of CT is no longer in doubt. Thus, the aim of this pragmatic real-life study was to assess the organizational impact and to find out whether the management of patients was improved by ultra-low-dose CT. Thus, we considered this to be the best way to evaluate diagnostic agreement in this retrospective study, and it appeared interesting to evaluate if imaging performed in ED could modify the management in ED when considering the hospital discharge as the reference. “
- Although CT is available in most emergency departments, the cost-effective benefit should also be considered. Is there a financial analysis available in this database?
Response: Medico-economic data have also been collected, but will be the subject of a separate study.
- How are the image results different from the discharge results?
Response: In the unweighted population, image results were concordant with discharge results for 838 (68.7%) patients and for 103 (88%) patients respectively in the radiography group and in ULD CT group (p<0.001). We added this information in the manuscript line 229.
- The writing needs improvement. (abstract line 29-32)
Response: We modified the abstract as follows:
“Chest radiography remains the most frequently used examination in the emergency departments (ED) for the diagnosis of community-acquired pneumonia (CAP), despite its poor diagnostic accuracy compared with ultra-low-dose (ULD) chest computed tomography (CT). However, while ULD CT appears to be an interesting alternative to radiography, its organizational impact in the ED remains unknown. »
Reviewer 2 Report
This is a retrospective study that included patients admitted to ED for CAP. The purpose of the authors was to evaluate the organizational benefits of representation. The study included 1609 patients (1476 had a chest x-ray and 133 had a CT). The idea is original and clinically useful, but in my opinion, the methodology of Agreement Analysis is incorrect. An OR was used (probably a Quadro test chi), but I would suggest an agreement analysis (coefficient Gwet AC1, Cohen kappa....).
Author Response
Reviewer 2
This is a retrospective study that included patients admitted to ED for CAP. The purpose of the authors was to evaluate the organizational benefits of representation. The study included 1609 patients (1476 had a chest x-ray and 133 had a CT). The idea is original and clinically useful, but in my opinion, the methodology of Agreement Analysis is incorrect. An OR was used (probably a Quadro test chi), but I would suggest an agreement analysis (coefficient Gwet AC1, Cohen kappa....).
Response:
We thank the reviewer for his pertinent comment. We agree that Cohen’s kappa or Gwet AC1’s coefficient are used to assess the degree of agreement between categorical variables. These two statistics are frequently used to test interrater reliability. However, this was not the goal of our diagnosis “agreement” analysis. The aim was to compare the rate of diagnostic agreement between the two groups (Chest radiography and ULD chest CT) and to assess the effect of imaging strategy on diagnostic agreement between ED discharge and hospitalization discharge. We performed a logistic regression on a weighted population to minimize confounding factors and to ensure the comparability of the baseline covariates between the two groups. The OR represent the strenght of the association between the imaging strategy and the probability of agreement. We explained this analysis in the methods section line 150.
Reviewer 3 Report
In summary, the article effectively presents the organizational advantages of using ultra-low-dose thoracic scans CT compared with chest radiography for the diagnostic workup of community-acquired pneumonia. The study design, statistical analysis, and clear presentation of results contribute to the overall strength and reliability of the findings.
Author Response
Reviewer 3
In summary, the article effectively presents the organizational advantages of using ultra-low-dose thoracic scans CT compared with chest radiography for the diagnostic workup of community-acquired pneumonia. The study design, statistical analysis, and clear presentation of results contribute to the overall strength and reliability of the findings.
Response: We thank the reviewer for his comment and for appreciating our work.
Round 2
Reviewer 2 Report
No further comments from my side
Author Response
We thank the reviewer for his comment.